# Single and Repeated Oral Dose Toxicity and Genotoxicity of the Leaves of Butterbur

**DOI:** 10.3390/foods10081963

**Published:** 2021-08-23

**Authors:** Sangsu Park, Jeongin Lim, Kyung Tae Lee, Myung Sook Oh, Dae Sik Jang

**Affiliations:** 1Department of Fundamental Pharmaceutical Sciences, Graduate School, Kyung Hee University, Seoul 02447, Korea; x-zara@nate.com; 2NATUREBIO Co., Ltd., Seoul Biohub Industry-Academic Cooperation Center, Seoul 02447, Korea; naturebio2020@nate.com; 3Department of Pharmaceutical Biochemistry, College of Pharmacy, Kyung Hee University, Seoul 02447, Korea; ktlee@khu.ac.kr; 4Department of Oriental Pharmaceutical Science, College of Pharmacy, Kyung Hee University, Seoul 02447, Korea; msohok@khu.ac.kr; 5Department of Life and Nanopharmaceutical Sciences, Graduate School, Kyung Hee University, Seoul 02447, Korea

**Keywords:** *Petasites japonicus* (Siebold & Zucc.) Maxim, oral toxicity test, genotoxicity test, safety assessment

## Abstract

Butterbur (*Petasites japonicus* (Siebold & Zucc.) Maxim) leaves are available to consumers in the marketplace, but there is no guarantee that they are safe for human consumption. Previously, we demonstrated that hot water extracts of *P. japonicus* leaves (KP-1) had anti-inflammatory properties and attenuated memory impairment. However, data regarding KP-1 toxicity are lacking. This study assessed the safety of KP-1 by examining oral and genotoxic effects using in vivo and in vitro tests, respectively. In a single oral dose toxicity and two-week repeated oral dose toxicity study, we observed no toxicologically significant clinical signs or changes in hematology, blood chemistry, and organ weights at any dose during the experiment. Following a thirteen-week repeated oral dose, toxicity, hyperkeratosis, and squamous cell hyperplasia of the limiting ridge in the stomach were observed. The no observable adverse effect level (NOAEL) was found to be 1250 mg/kg/day in male and female rats. However, hyperkeratosis and hyperplasia were not considered to be of toxicological significance when extrapolating the NOAEL to humans because the limiting ridge in the stomach is species-specific to rats. Therefore, in our study, the NOAEL was considered to be 5000 mg/kg/day when the changes in the stomach’s limiting ridge were discounted. Moreover, in vitro bacterial reverse mutations and chromosomal aberrations in Chinese hamster lung (CHL) cells and the in vivo micronucleus in Institute of cancer research (ICR) mice assays showed that KP-1 possessed no mutagenicity. Although additional research is required, these toxicological evaluations suggest that KP-1 could be safe for human consumption.

## 1. Introduction

Butterbur (*Petasites japonicus* (Siebold & Zucc.) Maxim) is a herb in the family of Asteraceae, which is distributed in East Asia [1]. It is cultivated in South Korea and Japan, where it is mainly used as a vegetable [2]. The roots of *Petasites* species have been used as a muscle relaxant and to treat gastrointestinal disease and asthma in Europe for more than 2000 years [3,4]. *P. japonicus* leaf (PL) inhibits allergies, obesity, and asthma, which are associated with chronic inflammatory diseases [5]. This anti-inflammatory effect of PL is associated with various types of phenolic compounds, such as fukinolic acid, petasinophenol, phenylprophenoyl, and sulfonic acid, which inhibit free radical scavenging and nitric oxide (NO) production in macrophages [6,7]. Quercetin and kaempferol 3-*O*-(6”-acetyl)-β-glucopyranoside from PL have a powerful inhibitory effect on β-secretase, and the sesquiterpenes from PL have neuroprotective effects [7,8]. Furthermore, we recently reported that an aryltetralin lactone lignan, petasitesin A, and a phenolic compound, cimicifugic acid D, from PL showed significant inhibitory effects on NO and prostaglandin E_2_ (PGE_2_) production in lipopolysaccharide (LPS)-stimulated macrophages [9]. Our previous study demonstrated the neuroprotective effects of hot water extracts of *P. japonicus* leaves (KP-1) in Aꞵ_25-35_-plaque-induced Alzheimer’s disease models. KP-1 protects hippocampal cells against Aꞵ_25-35_ plaque toxicity by regulating intracellular reactive oxygen species (ROS) levels and in vitro activation of cAMP response element-binding protein (CREB)-induced antioxidant enzymes [10]. Development of and research on pharmaceutical and functional foods using natural products are actively progressing, which reinforces the need for safety assessments. In particular, the assessment of toxicity is essential to safety assessments because some complex blends of natural products are toxic [11,12]. Thus, the present study was designed to obtain safety data by validating an analytical method for standardization and evaluating the in vivo single and repeated oral dose toxicity and in vitro genotoxicity of KP-1. This will be the basis for the future recognition of Butterbur as a health-functional food.

## 2. Materials and Methods

### 2.1. KP-1 Preparation and Calibration Standards and Quality Control Samples of Fukinolic Acid

Fresh PL (1000 kg) was purchased locally (Yeosu, Korea) and dried in a 50 °C air dryer for 24 h. The plant material was identified by one of the authors (D.S.J) and a plant specimen (PEJA-2019) was deposited at the Laboratory of Natural Product Medicine, College of Pharmacy, Kyung Hee University. The dried leaves (100 kg) were then boiled with a 20-fold quantity of distilled water at 100 °C for 4 h. The extract obtained was filtered, concentrated, and freeze-dried (Iisinbiobase, Gyeonggi, Korea) to yield 25 kg of powder. Then, the powder was blended with a 2-fold quantity of 70% ethanol at room temperature for 24 h. Its sediment was freeze-dried to give 20 kg of powder. The powder was stored at −20 °C until further use. A standard stock solution of the highest concentration was prepared by weighing 10 mg of the fukinolic acid (Naturebio, Seoul, Republic of Korea), placing it into a 10 mL volumetric flask, and diluting it with the dilution solution (concentration: 1000 µg/mL). Standard solutions were prepared by diluting this stock solution with the dilution solution, yielding final concentrations of 5–100 µg/mL.

### 2.2. Chromatographic Conditions

The high-performance liquid chromatography (HPLC) system consisted of a model LC-20AD pump, a model SIL-20AC autosampler, and a model SPD-20A (UV/VIS) detector set to 330 nm (Shimadzu, Kyoto, Japan). Separations were achieved at ambient temperature using an Ascentis^®^ C_18_ HPLC column (Supelco, St. Louis, MO, USA). Water and formic acid (Wako, Osaka, Japan) were mixed to 0.1% and used by mobile phase A. Acetonitrile (Burdick&Jackson, Muskegon, MI, USA) and formic acid were mixed to 0.1% and used by mobile phase B. The following gradient elution was used: 0 min, 5% B; 15 min, 15% B; 30 min, 40% B; 45 min, 95% B. The required amount (0.5 mL) of the dosing formulation at the concentration of 500 mg/mL was taken from each batch of dosing formulations and diluted with the solution for dilution by a factor of 2000, but the dosing formulation at the concentration of 0.25 mg/mL was not diluted. Both of the dosing formulations were sonicated for 30 min, filtered through a 0.45 µm syringe filter, and delivered at a rate of 0.8 mL/min. LC solution ver. 2.12 multi was used for process control and data collection.

### 2.3. Validation of the Analytical Method

#### 2.3.1. Specificity

The mobile phases, solution for dilution, and vehicle were analyzed to check for interfering peaks at the retention time of fukinolic acid to confirm the assay’s specificity. The result was judged to be acceptable when a sufficient number of peaks of the fukinolic acid was observed and there were no interfering peaks at the same retention time as the standard substance.

#### 2.3.2. Calibration Curves

The calibration solutions of concentrations from 5 to 100 µg/mL were analyzed. A correlation coefficient between the concentration and peak area of the standard solutions was generated. The calibration curve was used to evaluate the results of the stability analysis of the dosing formulations. The result was judged to be acceptable when the correlation coefficient (r) of the calibration curve was more than 0.9950 and the accuracy of each concentration was in the range of 85–115% of the theoretical values.

#### 2.3.3. Precision and Accuracy

Intra-day assay precisions were determined as relative standard deviations (RSDs), and intra-day assay accuracies were expressed as percentages of theoretical concentration, $\mathrm{as}\;\mathrm{accuracy}\;\left(\%,\;\right)=(\mathrm{mean}\;\mathrm{determined}\;\mathrm{concentration}/\mathrm{nominal}\;\mathrm{concentration}\times 100$. Precision was expressed as percentages of theoretical concentration as precision (%) = (standard deviation of determined concentrations/mean of determined concentration/nomical concentration) × 100. The samples from the middle layer of dosing formulations of each batch were collected in triplicate and analyzed once each. The result was considered to be acceptable when the precision was 10% or less and the accuracy was in the range of 80–120% of the theoretical values.

#### 2.3.4. Stability

The stock solution was allowed to stand at room temperature for 4 h and stored under refrigeration (2–8 °C) for 192 h. Samples were prepared in triplicate at the concentration of the QC sample and analyzed once each. The result was judged to be acceptable when the precision was 10% or less and the accuracy was in the range of 85%–115% of the theoretical values. All dosing formulations were stored at room temperature or under refrigeration and three replicate samples from the middle layer of dosing formulations of each batch were analyzed for stability. The results of intra-day variation were used as the result of the analysis performed immediately after preparation. The results were judged to be acceptable when the precision was 10% or less and the variation in stability was within ±15% of the initial concentration.

### 2.4. Animals

Sprague-Dawley rats (Orientbio Inc., Gyeonggi, Korea) were kept under a 12 h light/dark cycle (7 a.m. to 7 p.m. via an automated timer). Pelleted rodent chow (Envigo RMS Inc., Indianapolis, IN, USA) and public tap water (UV sterilized and filtered) were provided ad libitum. Animal rooms were acclimated to a constant temperature (21 ± 1 °C) and humidity (57.6 ± 7%). All animal handling procedures were reviewed and approved by the Institutional Animal Care and Use Committee (IACUC) of Biotoxtech Co., Ltd. based on the Animal Protection Act (Enactment 31 May 1991, No. 4379, Revision 20 March 2018, No. 15502) (Approval Nos. 180664, 180702, and 190012). All animals were sacrificed by exsanguination from the abdominal aorta under isoflurane anesthesia. Complete gross postmortem examinations were performed on all animals, including the external and internal surfaces.

ICR mice (Orientbio Inc., Gyeonggi, Korea) were kept under a 12 h light/dark cycle (7 a.m. to 7 p.m. via an automated timer), and pelleted rodent chow and public tap water were provided ad libitum. Animal rooms were acclimated to a constant temperature (21 ± 1 °C) and humidity (54.1 ± 7%). All animal handling procedures were reviewed and approved by the Institutional Animal Care and Use Committee (IACUC) of Biotoxtech Co., Ltd. based on the Animal Protection Act (Enactment 31 May 1991, No. 4379, Revision 20 March 2018, No. 15502) (Approval No. 180517). All animals were sacrificed by cervical vertebral dislocation at 24 h after the 2nd dosing of the KP-1.

### 2.5. The Assessment of Orally Acute Toxicity and Repeated Dose Toxicity

#### 2.5.1. Single Oral Dose Toxicity Study of KP-1

The single oral dose toxicity study was conducted following test guidelines by the Ministry of Food and Drug Safety (No. 2017-32) [13]. The animals included 12 males and 12 females, weighing 122.4–132.2 g and 110.9–122.8 g, respectively. Two groups (5 animals/sex/group) of 10 males and 10 females weighing 157.9–175.5 and 127.4–144.0 g, respectively, on the assignment day were selected and distributed. The dose-volume was set at 10 mL/kg/body weight. Individual doses were calculated for each animal based on the animal’s body weight recorded just prior to dosing. Animals were dosed via gastric intubation with a 3 mL disposable syringe fitted with an intubation tube. Animals were fasted approximately 16 h prior to dosing. In the dose range-finding study (*n* = 2), no dead animals were observed at 5000 mg/kg. Therefore, 5000 mg/kg was selected as the KP-1 group for this study. The negative control group was administered sterile distilled water (JW Pharmaceutical, Seoul, Korea) as a vehicle control.

#### 2.5.2. Two-Week Repeated Oral Dose Toxicity Study of KP-1 

The two-week oral dose toxicity study was conducted following test guidelines by the Ministry of Food and Drug Safety (No. 2017-71) [14]. Four groups (5 animals/sex/group) of 20 males and 20 females were selected and distributed. Animals were randomly assigned to groups to equalize mean group body weights. In the single oral dose toxicity study, no abnormal clinical signs or mortality were observed in the 5000 mg/10 mL/kg group. The high dose level was selected at 5000 mg/kg/day for this study. Then, the mid and low dose levels were selected at 2500 and 1250 mg/kg/day, respectively, by applying a geometric ratio of 2. The negative control group was administered sterile distilled water.

#### 2.5.3. Thirteen-Week Repeated Oral Dose Toxicity Study of KP-1

Four groups (10 animals/sex/group) of 40 males and 40 females were used for a repeated oral dose toxicity study for thirteen weeks. A dose of 5000 mg/kg/day was selected as the high dose level, and 2500 and 1250 mg/kg/day were selected as the mid and low dose levels, respectively, by applying a geometric ratio of 2 [15,16]. Control animals were dosed with the vehicle alone at the same amount as the KP-1 dosing groups. This assay was performed based on the guidelines by the Ministry of Food and Drug Safety (No. 2017-71) [14]. Body weights were recorded just prior to dosing on Day 1, once a week during the dosing period, and on the day of necropsy. Body weight data on the day of necropsy were not included in the evaluation of body weights since these data were body weights of fasting animals. Ophthalmological examination was conducted on both eyes of 5 males and 5 females in each group at thirteen weeks. The examination for the pupil light reflex and anterior segment of the eye was conducted with the naked eye before instillation of the mydriatic agent Isopto-atropine^®^ (Alcon, Fort Worth, TX, USA), and then the anterior segment of the eye, transparent media, and ocular fundus were observed with the naked eye and using an ophthalmoscope after instillation of the mydriatic agent. Fresh 3 h and 24 h urine samples were collected from 5 males and 5 females in each group at thirteen weeks. When fresh urine was collected, feeding and dosing were not performed. However, drinking water was provided ad libitum.

### 2.6. Genotoxicity Test 

#### 2.6.1. Bacterial Reverse Mutation Test

Five strains of Salmonella typhimurium (Tester strain for Ames (TA)98, TA1535, TA100, TA1537, and WP2uvrA (pKM101)) were purchased from Moltox Inc. (Boone, NC, USA). The plate incorporation test was performed both in the presence (0.5 mL of S9 mix) and in the absence (0.5 mL of 0.1 M phosphate buffer, pH 7.4) of metabolic activation in, briefly, an overnight bacterial culture (1 × 10^4^ cells/mL) and soft agar (0.5 mM L-Histidine/D-Biotin mixture at a ratio of 10 to 1 for Salmonella typhimurium and a 0.5 mM L-tryptophan solution at a ratio of 10 to 1 for Escherichia coli) [17]. The high dose was selected at 5000 μg/plate and it was sequentially diluted by applying a geometric ratio of 4 to produce lower dose levels (1250, 313, 78.1, 19.5, and 4.88 μg/plate). The growth inhibition by the KP-1 was evident at 5000 μg/plate in the TA100, TA1535, and TA1537 strains in the absence of metabolic activation. Growth inhibition by the KP-1 was not evident at any dose level in the TA98 and WP2uvrA (pKM101) strains in the absence of metabolic activation and in the TA98, TA100, TA1535, TA1537, and WP2uvrA (pKM101) strains in the presence of metabolic activation. Precipitation of the KP-1 was not evident at any dose level in any strain in the absence or presence of metabolic activation. Therefore, the dose levels of the main study were selected as shown in Table 1. In addition, the negative control group was administered sterile distilled water (JW Pharmaceutical, Seoul, Korea) and the positive group was administered Sodium azide, 2-Nitrofluorene, 2-Aminoanthracene, 9-Aminoacridine, and 4-Nitroquinoline N-oxide. All positive controls were purchased from Sigma-Aldrich^®^ (St. Louis, MO, USA).

#### 2.6.2. In Vivo Micronucleus Test in ICR Mice

Twenty-five male ICR mice were quarantined and acclimated for 7 days and observed once daily for clinical signs (dose range-finding study: 15 males and 15 females; main study: 25 males). The animals were moved to an animal room after they were acclimated in a quarantine room for 3 days. The high dose level of the main study was selected at 5000 mg/kg and it was sequentially diluted by applying a geometric ratio of 2 to produce lower dose levels (2500 and 1250 mg/kg). In addition, the positive group was administered Mitomycin C (Sigma-Aldrich, St. Louis, MO, USA), and the negative control group was administered sterile distilled water as a vehicle control group was set. Because no mortality was evident in either males or females in the dose range-finding study, only males, which are known to be susceptible to micronucleus induction, were used in the main study.

#### 2.6.3. In Vitro Mammalian Chromosomal Aberration Test

Clastogenic effects of KP-1 were analyzed in CHL/IU cells (Mpbiomedicals, Irvine, CA, USA). Thus, 5 × 10 ^4^ cells/mL were seeded in dishes. The high dose of the KP-1 was set at 5000 μg/mL. The high dose was sequentially diluted by applying a geometric ratio of 2 to produce lower dose levels (2500, 1250, 625, 313, 156, 78.1, 39.1, and 19.5 μg/mL). In addition, the negative control group was administered sterile distilled water and the positive group was administered 0.1 μg/mL Mitomycin C and 0.5 μg/mL benzo[a]pyrene (Sigma-Aldrich, St. Louis, MO, USA) was added as a positive control without or with S9 metabolic activation, respectively. After treatments, cells were exposed to colcemid (0.2 μg/mL final concentration) for 2 h of the incubation period. The metaphase cells were then harvested by a Trypsin-EDTA solution and fixed with Carnoy’s solution (methanol: acetic acid, 3:1). Three hundred cells in metaphase per dose were observed and scored. Chromosomal aberrations were classified into structural aberrations, numerical aberrations, and others.

### 2.7. Statistical Analysis

Statistical data analysis for body weight, food consumption, urine volume, hematology, clinical chemistry, organ weight, frequency of cells with chromosome aberrations, the incidence of micronucleated polychromatic erythrocytes (MNPCEs), and the ratio of polychromatic erythrocytes (PCE) to total erythrocytes was performed using version 9.3 of the SAS Program (SAS Institute Inc., Cary, NC, USA). The data were analyzed utilizing the Bartlett test for homogeneity of variance (*p*-value = 0.05). One-way analysis of variance (ANOVA) was employed on homogeneous data (*p*-value = 0.05); then, if significant, Dunnett’s test was applied for multiple comparisons (*p*-value = 0.05 and 0.01, two-tailed). The Kruskal–Wallis test was employed on heterogeneous data (*p*-value = 0.05); then, if significant, Steel’s test was applied for multiple comparisons (*p*-value = 0.05 and 0.01, two-tailed).

## 3. Results and Discussion

The industrial value of plant resources in the medicinal product market is prominent [18]. The ultimate standard for any drug is non-toxicity, efficacy, specificity, stability, and potency [19]. In that sense, natural products are more reliable resources for health-functional foods and medicines and supply effective drugs with fewer side effects and low toxicity [20]. Although herbal drugs are generally thought to be safe and effective, “natural” does not signify proof of safety [21,22]. There are many factors involved in determining the safety of a functional ingredient. These elements include compositional analysis, chemical structure analysis, and toxicity studies [23]. Safety evaluations are critical when new compounds/foods are introduced to the marketplace [24,25]. In our preliminary studies, we elucidated two new lignans and isolated six known compounds from KP-1 that showed neuroprotective and anti-inflammatory effects [9,10]. PL extracts are used as medicines in the West. On the other hand, PL is eaten as food in Asia [26]. The rhizomes and PL contain amounts of toxic pyrrolizidine alkaloids (PA), which have demonstrated carcinogenic and mutagenic potential [27]. Recent research indicated that PLs may cause adverse reactions in rats based on enhanced postprandial oxidative stress [28]. Additionally, a methanol extract of PL revealed hepatotoxicity [29]. Therefore, plants used for health-functional food and pharmaceutical purposes must be low in PA. KP-1 was extracted using 70% ethanol to reduce the amount of toxic PA in PL. PAs are soluble in organic solvents such as EtOH and MeOH and not stable at high temperatures [30,31]. Therefore, we removed the supernatant after 24 h by adding ethanol during the extraction [32]. Our results suggest that KP-1 could be useful for further toxicity evaluation due to their potential safety as a functional food. Therefore, we assessed potential toxic effects of KP-1. We acquired safety data to enable the development of KP-1 as a health-functional food. Consequently, validation of the KP-1 extraction method by HPLC, in vivo single and repeated oral dose toxicity tests, and in vitro genotoxicity tests were performed to assess the safety of KP-1.

### 3.1. Analytical Method and Validation of Fukinolic Acid in the KP-1 by HPLC

#### 3.1.1. HPLC Chromatogram and Specificity of Fukinolic Acid and KP-1

To validate the analytical method for fukinolic acid in the KP-1 dosing formulations, we used high-performance liquid chromatography (HPLC). The solvent conditions were established using Ascentis^®^ C_18_ (4.6 × 250 mm, 5 μm) with a detection wavelength of 330 nm [33]. The chromatogram of the fukinolic acid showed a sufficient number of peaks for the analysis, and chromatograms of the mobile phase, solutions for dilution, and vehicle did not reveal any interfering peaks in the standard substance (Figure 1).

#### 3.1.2. Linearity and Sensitivity

The standard solution was diluted stepwise to a concentration range of 5 to 100 µg/mL. The HPLC measurement value was prepared as a calibration curve. Under these analytical conditions, the calibration curve of the fukinolic acid had an average linear regression equation (*n* = 3) of y = 13,056.9148 (±137.6678) x + 14,860.9169 (±26,589.9388) (r = 0.9997 ± 0.0002). The accuracy of each concentration in the two calibrations was from 94.38% to 102.90% and showed high linearity.

#### 3.1.3. Precision and Accuracy of Intra-Day Variation in and Stability of Dosing Formulations

The precision results of the stock solution were 0.39% and 1.15% for the 0.02 mg/mL fukinolic acid, respectively. The accuracy results were 103.70% and 104.35%, respectively. At the completion of the analysis, three replicate 0.02 mg/mL QC samples were analyzed. The precision and accuracy results were 0.68% and 103.25% for 0 h and 0.61% and 98.95% for 192 h, respectively. The precision result of intra-day variation was 6.77% and 0.38% for the 0.25 and 500 mg/mL dosing formulations, respectively. The accuracy results were 105.86% and 117.53%, respectively, following 0 h of standing at room temperature. The precision result of the reanalysis of the 0.25 and 500 mg/mL dosing formulations to verify stability was 6.50% and 0.19%, respectively, following 4 h of standing at room temperature. The precision results of the reanalysis of the intra-day samples at the room temperature after 24 h were 9.36% and 0.63% for the 0.25 and 500 mg/mL dosing formulations, respectively. The precision results of the reanalysis of the 0.25 and 500 mg/mL dosing formulations to verify stability were 0.30% and 0.45%, respectively, following 192 h of storage under refrigeration (Table 2). Based on the results of the study, the 0.25 and 500 mg/mL dosing formulations in water for injection were stable for 4 h at room temperature and for 192 h under refrigeration.

#### 3.1.4. Homogeneity

The precision results on the homogeneity of the upper, middle, and lower layers were 6.36% and 0.45% for the 0.25 and 500 mg/mL dosing formulations, respectively. The accuracy results were 107.38% and 118.13%, respectively. Therefore, the precision and accuracy of the analysis of the homogeneity of the upper, middle, and lower layers of the 0.25 and 500 mg/mL dosing formulations satisfied the acceptance criteria (data not shown).

### 3.2. The Assessment of Oral Acute Toxicity and Repeated Dose Toxicity 

#### 3.2.1. Single Oral Dose Toxicity Study of KP-1

The purpose of this study was to assess the potential toxicity and to determine the approximate lethal dose of KP-1 following a single oral administration to six-week-old male and female Sprague-Dawley rats. Test groups consisted of a dose group at the dose level of 5000 mg/kg and a control group (injected with sterile water) with five animals of each sex per group. All animals were monitored for clinical signs and body weight changes during the 14-day observation period after dosing. They were euthanized and subjected to a gross necropsy at the end of the observation period. No animals died during the trial period and no experimental animal showed any unusual general clinical features (data not shown). Based on the results of this study, the approximate lethal dose of the KP-1 was greater than 5000 mg/kg in both male and female rats following a single oral administration under the conditions of this study.

#### 3.2.2. Two-Week Repeated Oral Dose Toxicity Study of KP-1

This study was conducted to assess the potential toxicity of KP-1 when administered once daily to Sprague-Dawley (SD) rats of both sexes by oral gavage for two weeks and to determine the dose levels for the repeated dose toxicity study. Three groups for KP-1 were designated at dose levels of 1250, 2500, and 5000 mg/kg/day in addition to a control group (using sterile water for the injection). Each group consisted of five males and five females. There were no deaths of animals in the 1250, 2500, and 5000 mg/kg/day groups during the study period. Regarding clinical signs, compound-colored excrement was observed in both sexes in the 1250, 2500, and 5000 mg/kg/day groups (Table 3). It was considered not to be toxicologically significant since KP-1-related changes were not observed in any other parameters [29]. There were no KP-1-related effects in body weight, food consumption, hematology, clinical chemistry, organ weights, and necropsy findings in males and females in the 1250, 2500, and 5000 mg/kg/day groups (data not shown). In conclusion, based on the result of the two-week toxicity study of KP-1 following oral administration to male and female rats, it was determined that the dose level for the high dose should be selected at 5000 mg/kg/day in the thirteen-week repeated dose toxicity study.

#### 3.2.3. Thirteen-Week Repeated Oral Dose Toxicity Study of KP-1

This study was conducted to evaluate the potential toxicity and safety of the KP-1 when administered by oral gavage once daily to Sprague-Dawley (SD) rats of both sexes for thirteen weeks. Three groups for the KP-1 were designated at dose levels of 1250, 2500, and 5000 mg/kg/day in addition to a control group (using sterile water for the injection). Each group consisted of 10 males and 10 females. During the observation period, observation of clinical signs, body weight, and food intake, ophthalmological examinations, and urinalysis were performed and, after the observation period, hematology, clinical chemistry, organ weights, gross post-mortem, and histopathological examinations were performed. During the dosing period, no deaths were observed in either sex in the 1250, 2500, and 5000 mg/kg/day groups, and compound-colored excrement and salivation before and/or after dosing were observed. The compound-colored excrement observed in this study was considered to be due to administration of an excessive amount of the test substance and the salivation before and/or after dosing was considered to be a change caused by the physicochemical properties of the test substance. In previous studies, salivation was observed in both sexes both before and after dosing. This is a common phenomenon due to the ingestion of food, which is not toxicologically significant [34]. No test-substance-related toxic effects were noted in body weights or food intake in males and females in the 1250, 2500, and 5000 mg/kg/day groups (Appendix A). There were no ocular abnormalities and other changes observed in the urinalysis were minor or incidental changes. Therefore, they were not considered to be of toxicological significance (Appendix A). There were no test-substance-related effects on hematology, clinical chemistry, or organ weights and minor changes in them were observed within the normal range. Accordingly, they were considered to not be of toxicological significance (Appendix A). At necropsy, yellow discoloration of the limiting ridge, thickening, and white discoloration of the fundic region in the stomach were observed in males in the 1250 mg/kg/day group and both sexes in the 2500 and 5000 mg/kg/day groups. A previous study reported that thickening of the walls in the stomach was observed in four males and five females and thickening in the limiting ridge in the stomach was observed in two males. Hyperplasia of squamous cells on the limiting ridge of the stomach was also observed in both sexes. However, this change was considered to be an incidental finding [34]. Based on the results of this study, the no observed adverse effect level (NOAEL) of KP-1 was considered to be less than 1250 mg/kg/day in males and to be 1250 mg/kg/day in female rats after repeated oral administration for thirteen weeks under the conditions of this study. However, these values were not considered to be of toxicological significance when extrapolating the NOAEL to humans because a limiting ridge in the stomach is an organ that exists only in rats [35,36,37]. Consequently, the NOAEL was considered to be 5000 mg/kg/day except for changes in the limiting ridge.

### 3.3. Genotoxicity Test

#### 3.3.1. In Vitro Bacterial Reverse Mutation Test

PL is an ancient medicinal plant that has been used worldwide for centuries. However, modern scientific studies have shown that its extracts contain toxic alkaloid constituents that are mutagenic and cancer-causing [38,39]. This study was designed to evaluate the mutagenic potential of KP-1 using histidine-requiring *Salmonella typhimurium* (TA98, TA100, TA1535, and TA1537) strains and the tryptophan-requiring *Escherichia coli* (WP2*uvrA* (pKM101)) strain in the absence and presence of metabolic activation. The mean number of revertant colonies was less than twice that in the negative control group at all dose levels of the KP-1 in all strains in the absence and presence of metabolic activation. In the positive control group, the mean number of revertant colonies for each strain was markedly increased (more than 2-fold) when compared with that of the negative control group (Figure 2). The KP-1 did not show any indication of mutagenic potential under the conditions of this study.

#### 3.3.2. In Vivo Micronucleus Test in ICR Mice

This study was designed to evaluate the potential of KP-1 to induce micronuclei in bone marrow cells of mice when orally administered via gastric intubation twice at 24 h intervals. The incidence of micronucleated polychromatic erythrocytes (MNPCEs) in polychromatic erythrocytes (PCEs) in the KP-1 groups was not significantly different from the negative control group (Appendix A). In addition, the ratio of PCEs to total erythrocytes in the KP-1 groups was not significantly different from the negative control group. In the positive control group, the incidence of MNPCEs in PCEs was significantly increased when compared with the negative control group. The ratio of PCEs to total erythrocytes in the positive control group was not significantly different from the negative control group (Appendix A).

#### 3.3.3. In Vitro Mammalian Chromosomal Aberration Test

This study was designed to evaluate the potential of KP-1 to induce chromosomal aberrations in Chinese Hamster Lung (CHL/IU) cells. The frequency of cells with chromosome aberrations in the short-term treatments with and without metabolic activation and in the continuous treatment without metabolic activation was not statistically significantly different when compared with the negative control group (Appendix A). In the positive control group, the frequency of cells with structural chromosome aberrations was statistically significantly increased when compared with the negative control group (Appendix A). KP-1 did not show any indication that it induces chromosome aberrations under the conditions of this study.

## 4. Conclusions

The HPLC method for the analysis of the dosing formulations of fukinolic acid in KP-1 was validated in this study. The 0.25 and 500 mg/mL dosing formulations injected using sterile water were homogeneous and stable for 4 h at room temperature and 192 h under refrigeration. In the single oral dose toxicity study, the approximate lethal dose of KP-1 was greater than 5000 mg/kg in both male and female SD rats. As a result of repeated oral administration for thirteen weeks of KP-1 to rats, hyperkeratosis and squamous cell hyperplasia of the limiting ridge in the stomach were observed in males in the 1250 mg/kg/day group and in males and females in the 2500 and 5000 mg/kg/day groups, respectively. Based on the results of this study, the no observed adverse effect level (NOAEL) of KP-1 was considered to be less than 1250 mg/kg/day in males and to be 1250 mg/kg/day in females. However, the hyperkeratosis and squamous cell hyperplasia were not considered to be of toxicological significance when extrapolating the NOAEL to humans because a limiting ridge in the stomach is species-specific to rats. Therefore, the NOAEL was considered to be 5000 mg/kg/day except for changes in the limiting ridge. The result of the Ames *Salmonella* assay, the in vivo micronucleus assay, and the chromosome aberration test showed that the KP-1 did not exert a genotoxic effect.

## Figures and Tables

**Figure 1 foods-10-01963-f001:**
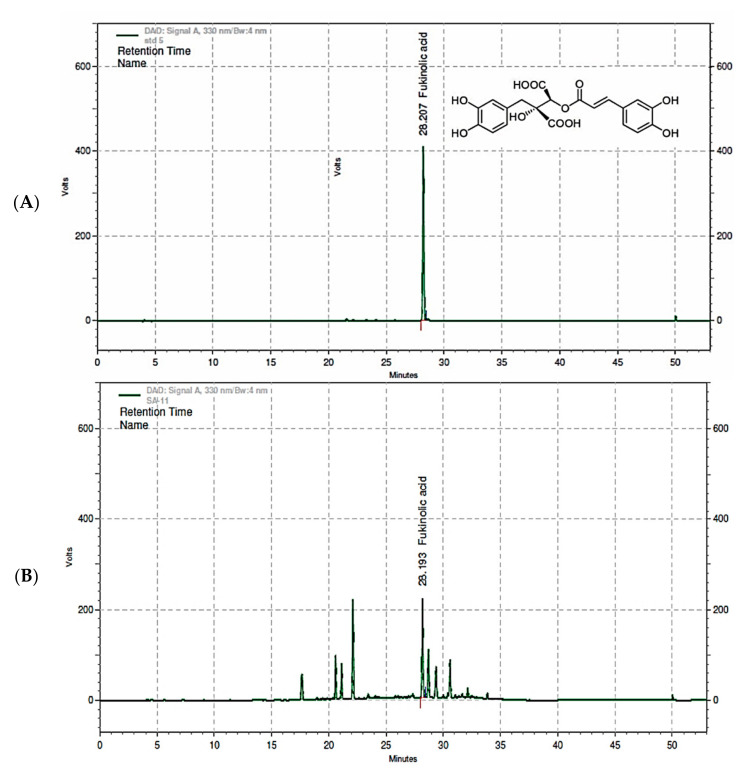
The HPLC chromatogram of fukinolic acid as the standard (**A**) and the *Petasites japonicus* leaf extract (KP-1) (**B**).

**Figure 2 foods-10-01963-f002:**
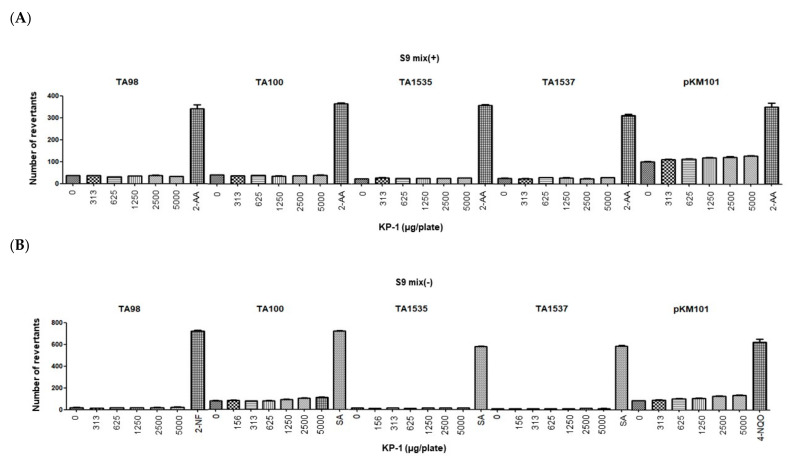
Effect of KP-1 on bacterial reverse mutation (**A**) with (+S9 mix) or (**B**) without (-S9 mix) the metabolic activation of *Salmonella tryphiurium* (TA98, TA100, TA1535, TA1537) and *Escherichia coli* (pKM101). A positive bacterial reverse mutation was induced by the positive control. 2-AA, 2-Amonoanthracene; 2-NF, 2-Nitrofluorene; SA, sodium azide; 4-NQO, 4-Nitroquinoline N-oxide.

**Table 1 foods-10-01963-t001:** Dose Levels of the Main Study.

Strain	S9 Mix	Dose Levels of the Main Study (µg/Plate)
TA98, WP2*uvrA* (pKM101)	−/+	5000, 2500, 1250, 625, 313
TA100, TA1535, TA1537	−	5000, 2500, 1250, 625, 313, 156
	+	5000, 2500, 1250, 625, 313

**Table 2 foods-10-01963-t002:** Stability of Fukinolic acid and KP-1.

Substance	Temperature	Hours	Concentration (mg/mL)	Precision (%)	Accuracy (%)
Fukinolic acid	room	0 h	0.02 ^1^	0.39	103.70
			0.02 ^2^	0.68	103.25
	refrigeration	192 h	0.02 ^1^	1.15	104.35
			0.02 ^2^	0.61	98.95
KP-1	room	0 h	0.25	6.77	105.86
			500	0.38	117.53
		4 h	0.25	6.50	92.35
			500	0.19	117.61
		24 h	0.25	9.36	97.58
			500	0.63	119.56
	refrigeration	192 h	0.25	0.30	114.11
			500	0.45	117.53

^1^ concentration of the standard solution; ^2^ concentration of the quality control (QC) sample.

**Table 3 foods-10-01963-t003:** Summary of Clinical Signs and Necropsy Findings.

**Sex: Male**			
**Group/Dose (mg/kg/day)**	**No. of Animals**	**No. of Clinical Sign Animals** **(Compound-Colored Excrement)**	**NOA ^1^**
G1/0	5	0	5
G2/1250	5	4	5
G3/2500	5	4	5
G4/5000	5	5	5
**Sex: Female**			
**Group/Dose (mg/kg/day)**	**No. of Animals**	**No. of Clinical Sign Animals** **(Compound-Colored Excrement)**	**NOA ^1^**
G1/0	5	0	5
G2/1250	5	3	5
G3/2500	5	4	5
G4/5000	5	5	5

^1^ NOA: No Observable Abnormality.

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
