# Peer review of "Single and Repeated Oral Dose Toxicity and Genotoxicity of the Leaves of Butterbur"

_foods, 2021, doi:10.3390/foods10081963_

Round 1
Reviewer 1 Report
Phrases like: "polyphenol-fukinolic acid compounds" (L42) and "the flavonoid of PL" (L45) are not precise and should be corrected. Lines 52/53 - pause shoud be removed. Paragraph 2.1 descriptions are unclear - do percentage numbers (L 68&70) refer to 1 ton of fresh plant material? Line 96; please correct "peaksat". Is "acclimation" (Lines 151, 163, 250) a correct word? Line 233 - please correct to benzo[a]pyrene. Lines 302 and 303 should be united.
Author Response
We appreciate the reviewer’s valuable comments and the opportunity to re-submit the manuscript, entitled “Single and Repeated Oral Dose Toxicity and Genotoxicity of the Leaves of Butterbur”. The reviewer’s comments are insightful and we have revised the manuscript accordingly.

Reviewer 2 Report
Please find the attachment.

Author Response

(The authors gave the same response as above.)

Round 2
Reviewer 2 Report
The revised manuscript addresses all the points raised by peer reviewers, and corrections have been incorporated in the revised manuscript accordingly.
Author Response
We appreciate the editor’s valuable comments and the opportunity to re-submit the manuscript, entitled “Single and Repeated Oral Dose Toxicity and Genotoxicity of the Leaves of Butterbur”.
Below, we are reflected with highlight in the revised manuscript in the attached file.
